# OpenReview forum: "Sponge Tool Attack: Stealthy Denial-of-Efficiency against Tool-Augmented Agentic Reasoning"
_ICML.cc/2026/Conference — ICML 2026 regular_

### Official Review · Reviewer_JqPg · 2026-03-11

**Soundness:** 2
**Presentation:** 3
**Significance:** 2
**Originality:** 2
**Overall Recommendation:** 4
**Confidence:** 2

**Summary:**

The paper introduces the "Sponge Tool Attack" (STA), a Denial-of-Efficiency (DoE) attack against tool-augmented LLM agents. Using a "query-only" threat model, STA rewrites input prompts to be convoluted yet semantically unchanged. This forces the victim agent into redundant tool-calling cycles, significantly inflating computational and time costs.  To generate these prompts, STA uses an offline multi-agent framework to distill reusable rewriting policies. Extensive evaluations across various LLMs, tools, frameworks, and datasets demonstrate the attack's broad effectiveness

**Compliance With Llm Reviewing Policy:**

Affirmed.

**Final Justification:**

I thank the authors for their efforts. I'll maintain my rating as 4.

**Key Questions For Authors:**

I have the following questions:
[I] Can the authors include some experiments towards defenses, either as ablation studies or preliminary experiments in some capacity?
[II] Some samples can be validated with a human in the loop to further make the claims more confident.

**Limitations:**

Yes

**Strengths And Weaknesses:**

Strengths
[I] The work has considerable novelty and seems highly relevant to me, as Denial-of-Efficiency (DoE) via resource exhaustion in agentic tool-calling is an underexplored area of work.
[II] The attack correctly avoids assuming gradient access or internal system configurations, constraining the adversary strictly to read-only query interaction. This black-box approach faithfully reflects real-world API deployment scenarios.
[III] The experimental scale is highly commendable. They have tested the attack across models of varying sizes and families.

Weaknesses:
[I] The paper focuses heavily on the attack but offers very little in terms of robust defenses or system-level mitigation strategies. Exploring simple heuristics or dynamic tool-budgeting would make the paper more well-rounded
[II] The paper claims the rewrites are "benign-looking" and maintain user intent to remain stealthy. However, there is no human-in-the-loop evaluation to confirm whether system moderators or actual users would flag these convoluted prompts as anomalous.

---

> ### Author Rebuttal · Authors · 2026-03-30
>
> It’s our great honor to receive Reviewer JqPg's thoughtful comments and kind support for our work. We would like to address the concerns as below.
>
> ---
>
> > **[W1,Q1]** Suggestion to include experiments on potential defenses.
>
> **A1:** Thanks for the valuable suggestion. To further evaluate the attack performance and stealthiness of STA, we compare STA with an explicit prompt-injection method (ThinkTrap [1]). Moreover, for defense, we additionally apply **Prompt-Guard-86M**, a guardrail model designed to detect jailbreak or prompt injection in input prompts, and report the detection AUC. Results are shown below (Qwen2-7B, Low-Budg. setting).
>
> |Method|ΔSteps↑|\|Similarity\|↓|Guardrail AUC↓|
> |-|-|-|-|
> |ThinkTrap [1]|2.31±0.83|2.12±0.47|0.874|
> |**STA (ours)**|**3.33±0.99**|**0.70±0.20**|**0.103**|
>
> As can be observed, ThinkTrap increases reasoning steps but remains weaker than STA. Moreover, its **explicit encouragement of prompt complexity** significantly degrades semantic quality, making the attack **less stealthy and easier to detect**.
>
> We sincerely hope that our responses adequately address the concern and we will include the discussion in our revision.
>
> [1] ThinkTrap: Denial-of-Service Attacks against Black-box LLM Services via Infinite Thinking
>
> ---
>
> > **[W2,Q2]** Further evaluation with human-in-the-loop.
>
>
> **A2:** Thanks for highlighting this important consideration. Here, we additionally conduct **human evaluation** and **LLM-as-a-judge analysis**. Specifically, we randomly sample 100 pairs of original and rewritten prompts from our experimental data (with Gemma-27B as the vicitm model). Each pair is independently rated for semantic similarity on a 1–5 Likert scale by **five human experts as well as GPT-5.4**. We then compute sample-level rank correlations (**Spearman’s ρ**) between these two scores and our embedding-based similarity score.
>
> |Metric|Human vs. Embedding|LLM vs. Embedding|Human vs. LLM|Human IAA|
> |-|-|-|-|-|
> |Spearman’s ρ|**0.73**|**0.75**|**0.79**|**0.84**|
>
> These results suggest that the embedding-based similarity provides **a reliable proxy for semantic preservation**. We will include this additional analysis and discussion in the revision.
>
> ---
>
> Thanks again for the thoughtful and constructive feedback. If reviewer JqPg has any remaining concerns, we would definitely love to clarify further.

---

> > ### Author Rebuttal · Reviewer_JqPg · 2026-04-03
> >
> > Thank you for addressing my questions; my concerns have been adequately addressed, given the time constraints.

---

> > > ### Author Response · Authors · 2026-04-03
> > >
> > > Thanks to Reviewer JqPg for acknowledging that our response has adequately addressed the concerns, and we sincerely appreciate Reviewer JqPg’s strong support for our work.
> > >
> > > Thanks again to Reviewer JqPg for the valuable and constructive feedback throughout the review process.

---

### Official Review · Reviewer_7R3h · 2026-03-12

**Soundness:** 3
**Presentation:** 3
**Significance:** 3
**Originality:** 3
**Overall Recommendation:** 4
**Confidence:** 3

**Summary:**

The paper studies a new attack surface in tool-augmented LLM agents called Denial-of-Efficiency (DoE). The authors introduce Sponge Tool Attack (STA), a query rewriting framework that increases the number of tool-calling steps while preserving the original task semantics. STA constructs an offline policy bank using LLM-based rewriting and judging, then applies the learned policies to inflate tool usage during inference.

The evaluation spans six models, twelve tools, four agent frameworks, and thirteen datasets. Results show that rewritten queries consistently increase the number of tool calls across models and frameworks. The work highlights a potential economic vulnerability in tool-augmented agents where adversarial inputs increase execution cost without obviously degrading task quality.

**Compliance With Llm Reviewing Policy:**

Affirmed.

**Key Questions For Authors:**

1. How does STA compare with simple baselines such as appending instructions that explicitly request verification or multiple reasoning paths? Please include quantitative comparisons.

2. Can you provide a concrete sceneario where these would be an effective attack. I.e. attacker use the service but doesn't pay for llm calls?

3. What is the approximate computational cost of constructing the policy bank and running the attack? How does this compare to the additional cost inflicted on the victim system?

**Limitations:**

The Impact Statement (page 9) is brief and framed mostly positively, emphasizing safer agentic framework design. This is incomplete.

It does not acknowledge the dual-use risk of releasing a fully reproducible attack framework with prompts and policies that could be directly deployed against real systems. It also does not discuss the cost asymmetry of the attack, which may disproportionately affect smaller providers.

**Strengths And Weaknesses:**

### Strengths

- Introduces a plausible and underexplored attack surface in tool-augmented agents centered on efficiency degradation rather than correctness failure.
- The offline policy bank design is practical and clearly separates attack construction from deployment.
- Evaluation breadth is strong, covering multiple models, frameworks, tools, and datasets.
- The transferability of policies across victim models is an interesting empirical finding.

### Weaknesses

- **Missing attack baselines (critical).** The experiments only compare original queries with STA. A trivial baseline is absent. For example, simply appending an instruction such as "verify the answer step by step using multiple independent approaches" could plausibly increase tool usage. Several examples in Appendix H resemble this pattern. If such a simple modification produces similar inflation, the necessity of the full STA pipeline becomes questionable.

- **Threat model is underwhelming.** For the attack to matter economically, several assumptions must hold: the service provider pays for tool calls, the attacker can observe tool-call counts during probing, rate limiting or anomaly detection is absent during attack search. The paper should clarify which deployment scenarios realistically satisfy these assumptions.

- **Related work omissions.** Some closely related prior art are not discussed:

  - Denial-of-Service Poisoning Attacks against Large Language Models.
    https://arxiv.org/abs/2410.10760
  - Crabs: Consuming Resource via Auto-generation for LLM-DoS Attack under Black-box Settings.
    https://arxiv.org/abs/2412.13879
  - OverThink: Slowdown Attacks on Reasoning LLMs.
    https://arxiv.org/abs/2502.02542

  These papers target related forms of resource amplification or slowdown in LLM systems. They should at least be discussed in the related work section.



- **Attack construction cost not reported.** Building the policy bank requires repeated rewriting and judging rounds plus victim agent executions. The paper does not quantify the computational cost of generating attack policies. Given the modest step inflation reported, the attack cost to damage ratio should be analyzed.

---

> ### Author Rebuttal · Authors · 2026-03-30
>
> It’s our great honor to receive Reviewer 7R3h's thoughtful comments and kind support for our work. We would like to response to the questions as below.
>
> ---
>
> > **[W1,Q1]** Compare with simple baselines.
>
> **A1:** Thanks for the valuable comment. We introduce a prompt-injection (PI) baseline, where the text blow is appended to the original query:
>
> - "Please reason thoroughly: explore multiple paths and perform extra checks before answering."
>
> The results are as below (Gemma-3-27B, Low-Budg.):
>
> Setting|ΔSteps ↑|\|Similarity\| ↓|Δ Cap Hits (%) ↑
> -|-|-|-
> PI Baseline|0.47±0.14|0.52±0.19|0.91
> Full STA|1.36±0.50|0.87±0.23|3.83
>
> The baseline slightly increases reasoning steps but remains much weaker. Unlike the surface-level redundancy introduced by the baseline, STA induces structured inefficiency (e.g., targeting stalling subgoals or hard-to-verify options), yielding more pronounced effects.
>
> ---
>
> > **[W2,Q2]** Regarding the threat model.
>
> **A2:** Thanks for the comment. We respond point by point, including (i) real-world scenarios, (ii) the attacker’s ability to observe certain information, and (iii) performance against potential defenses.
>
> **(i) Realistic deployment scenarios exist.** Some services offer **free hosted agents** (e.g., flight-booking assistants like Trip.com’s *TripGenie*), where the provider bears the operational cost. Also, in such case, **rate limiting is unlikely to mitigate our attack**, since practical limits typically **constrain queries** (queries per day) rather than the **token usage within each query** (tokens per query).
>
> **(ii) Observing tool-call counts is not a must.** **First**, our experiments show **strong cross-model transferability (Fig. 6(b))**, enabling a **surrogate threat model** where policies are constructed on a trace-visible agent (Fig.2) and then transferred to an answer-only target (Fig.3). **Second**, when tool-call counts are unavailable, we can consider **execution latency as a proxy signal** (since additional tool call incurs extra latency) by replacing the step-induction score with $ r_{\text{DoE}}^{\text{t}}(q)=5\cdot\sigma\left(\frac{T(q)-T_o}{T_o}\right)$, where T(q) is the execution time of the rewritten prompt, and $T_o$ is that of the original query. The sigmoid normalization $\sigma$ keeps the reward range consistent with Eq.(8). In the table below (Low Budg.) we can observe that despite slight degradation, the attack remains effective.
>
> Metric|Qwen2-VL-7B|GPT-4o-mini
> -|-|-
> ΔSteps↑|2.47±0.84|1.84±0.27
> \|Similarity\|↓|0.92±0.32|0.93±0.16
>
> **(iii) Attack effectiveness under defense.** STA avoids explicit prompt injection through semantic-preserving rewrites, making it difficult for defenses to detect. Here we compare STA with a recent prompt-injection method (ThinkTrap [1]) and evaluate the detection AUC using **Prompt-Guard-86M** (Qwen2-7B, Low-Budg.).
>
> Method|ΔSteps↑|\|Similarity\|↓|Guardrail AUC↓
> -|-|-|-
> ThinkTrap [1]|2.31±0.83|2.12±0.47|0.874
> **STA (ours)**|**3.33±0.99**|**0.70±0.20**|**0.103**
>
> ThinkTrap increases reasoning steps but remains weaker than STA. Moreover, its **explicit encouragement of prompt complexity** significantly degrades semantic quality, making the attack less stealthy and easier to detect.
>
> [1] ThinkTrap: Denial-of-Service Attacks against Black-box LLM Services via Infinite Thinking
>
> ---
>
> > **[W3]** Discuss related works on language (reasoning) models.
>
> **A3:** Thanks for the suggestion. We will **revise the Related Work Section** and add a new subsection titled **"Resource Amplification on Language (Reasoning) Models"** to include these papers (P-DoS, Crabs, OverThink).
>
> ---
>
> > **[W4]** Report the attack construction cost.
>
> **A4:** Thanks for the valuable suggestion. Using GPT-4o-mini pricing, the two tables below report: **(i)** the per-sample attack amplification, and **(ii)** a dataset-level comparison between **attack amplification** and **policy construction cost** (Cost Ratio = attack amplification / policy cost).
>
> - The per sample attack amplification:
>
> Dataset|Query Tokens (Orig.)|Query Tokens (Attack)|Completion Tokens (Orig.)|Completion Tokens (Attack)|USD Cost
> -|-|-|-|-|-
> VQAv2|~120k|~403k|~3k|~11k|**3.4×**
> SciFiBench|~81k|~231k|~3k|~7k|**2.8×**
> CLEVR-Math|~134k|~274k|~5k|~13k|**2.1×**
>
> - attack amplification vs. policy construction cost
>
> Dataset|VQAv2|SciFiBench|CLEVR-Math
> -|-|-|-
> Cost Ratio ↑|**25.7×**|**21.1×**|**16.6×**
>
> We observe that the attack increases both query and completion tokens per sample (USD Cost increase \~2.0×–3.0×), and the attack amplification greatly exceeds the policy construction cost (\~20×). We will include these results in the revision.
>
> ---
>
> > **[L1]** Further complete the Impact Statement.
>
> **A4:** Thanks for the comment. We will revise the Impact Statement to explicitly discuss the dual-use risks and the cost asymmetry that may affect smaller providers.
>
> ---
>
> Thanks again for the thoughtful and constructive feedback. We are happy to clarify any remaining concerns from reviewer 7R3h.

---

> > ### Author Rebuttal · Reviewer_7R3h · 2026-04-02
> >
> > I don't have any further questions. I keep weak-accept.

---

> > > ### Author Response · Authors · 2026-04-02
> > >
> > > We are very grateful to hear that our response has well addressed all the questions of Reviewer 7R3h, and we sincerely appreciate Reviewer 7R3h's strong support for our work.
> > >
> > > Thanks again to Reviewer 7R3h for the insightful and constructive feedback throughout the review process.

---

### Official Review · Reviewer_x2yo · 2026-03-13

**Soundness:** 2
**Presentation:** 3
**Significance:** 3
**Originality:** 3
**Overall Recommendation:** 4
**Confidence:** 3

**Summary:**

This paper introduces Sponge Tool Attack (STA), a black-box, prompt-rewriting attack that targets tool-augmented LLM agents to induce Denial-of-Efficiency (DoE). Under a strict, non-invasive setting, STA rewrites user prompts to preserve task semantics while reliably increasing the number of tool calls and intermediate reasoning steps, thereby inflating computational cost. The method builds an offline “policy bank” via a multi-agent loop (rewriter, judge, inductor) from a small probe set and then performs single-step, query-aware rewrites at deployment; extensive experiments across six models, four frameworks, twelve tools, and thirteen datasets show consistent step amplification with limited accuracy degradation.

**Compliance With Llm Reviewing Policy:**

Affirmed.

**Final Justification:**

I am keeping my score at 4 because I believe the paper makes a meaningful contribution, but the remaining concerns are substantial enough to prevent a stronger endorsement. The rebuttal improves the paper by addressing questions about trace-free applicability, semantic preservation, real cost inflation, and additional baselines, but these additions do not fully remove my concerns about evaluation depth and threat-model realism in the original submission. As a result, I view the work as above the acceptance bar, but only narrowly so, which is best reflected by a Weak Accept.

**Key Questions For Authors:**

Q1. How does STA perform if the adversary cannot observe intermediate traces (τ) or step counts during the offline policy construction phase?

Q2. Can you provide a breakdown of actual API cost inflation? For example, in a gpt-4o-mini setting, what is the average percentage increase in USD per query under attack?

Q3. How does STA compare against a "Zero-Shot" baseline where an LLM is simply asked to "rewrite this query to be as complex as possible"?

Q4. Are the policies overfitting to a particular framework’s tool interfaces or traces? How do results change if explicit tool names are hidden from LLM components inside STA?

**Limitations:**

No. The paper includes only a very brief impact statement saying the attack surface can guide safer framework design and motivate monitoring, but it does not meaningfully discuss misuse risks, deployment harms, or the method’s own limitations in enough depth.

The authors should add a short but concrete discussion of negative societal impact, including how STA could be used to raise inference cost, increase latency, exhaust tool budgets, and degrade service reliability for real users of agentic systems.

**Strengths And Weaknesses:**

Strengths :

1. The formalization of Denial-of-Efficiency (DoE) is a significant contribution, shifting the focus from "safety" (e.g., jailbreaking) to "operational cost" and "resource exhaustion" in agentic systems

2.  The attack assumes a strict query-only/read-only access constraint. By avoiding gradient access or internal model modifications, the authors demonstrate a highly realistic threat to publicly accessible agent APIs

3. The work demonstrates that policies induced from one model (e.g., gpt-4o-mini) can effectively attack others (e.g., Qwen2-VL-7B), suggesting that the attack captures model-agnostic reasoning vulnerabilities.

Weakness:

1.  There is an inconsistency between the abstract’s claim of "strict query-only access" and the methodological requirement for read-only access to internal information, specifically tool-calling traces and step counts. While step counts may be visible in some developer logs, many commercial "black-box" agents only return a final answer, which would render the current probe-and-reward loop (Eq. 6-7) difficult to implement.

2. The semantic-preservation penalty relies on a single embedding-based similarity; no human evaluation or task-specific equivalence checks are reported, raising questions about true stealth (especially for multi-turn, tool-heavy reasoning).

3. The primary metric for DoE is the increase in tool-calling steps (ΔSteps). However, the paper does not quantify the actual monetary cost, latency (wall-clock time), or token consumption. Since different tools (e.g., a simple calculator vs. an intensive web search) have vastly different costs, "more steps" does not always translate to a proportional "sponge" effect on resources.

4. The text mentions a range of [−5,0] for r smt  , but the linear mapping used for rewards requires careful calibration to avoid stepfulness (r
DoE) overwhelming semantic preservation

5. Excluding samples that already hit the tool cap biases the reported ΔSteps upward; include them in aggregate cap-hit metrics to avoid selection bias.

6. Limited discussion and no empirical comparison against inference-time DoS/DoE work focused on length amplification or “infinite thinking” (e.g., ThinkTrap; ReasoningBomb). While STA targets tools rather than purely token generation, these are closely related efficiency attacks and merit direct comparison and positioning.
Limited coverage of indirect prompt injection against agents that manipulate tool usage stealthily (e.g., adaptive IPI methods).

---

> ### Author Rebuttal · Authors · 2026-03-30
>
> It’s our great honor to receive Reviewer x2yo's thoughtful comments and kind support for our work. We would like to address the concerns as below.
>
> ---
>
> > **[W1,Q1]** Regarding the query-only access.
>
> **A1:** Thanks for the important comment. We would like to clarify this in two points. **First**, in our experiments, **Fig.6(b) shows strong cross-model transferability**. This enables a **surrogate threat model** where policies are conducted on a trace-visible agent and transferred to an answer-only target.
>
> **Second**, when traces or step counts are unavailable, we can consider **execution latency as a proxy signal** (since additional tool call typically incurs extra latency) by replacing the step-induction score with $ r_{\text{DoE}}^{\text{t}}(q)=5\cdot\sigma\left(\frac{T(q)-T_{o}}{T_{o}}\right)$, where T(q) is the execution time of the rewritten prompt, and $T_o$ is that of the original query. The sigmoid normalization $\sigma$ keeps the reward range consistent with Eq.(8). In the table below we can observe that despite slight degradation, the attack remains effective.
>
> |Metric|Qwen2-VL-7B|GPT-4o-mini
> |-|-|-
> |ΔSteps↑|2.47±0.84|1.84±0.27
> |\|Similarity\|↓|0.92±0.32|0.93±0.16
>
> ---
>
> > **[W2]** Question on the semantic-preservation penalty.
>
> **A2:** Thanks for raising this point. Here we further conduct **human evaluation** and **LLM-as-a-judge analysis**. We randomly sample 100 pairs of original and rewritten prompts (Gemma-27B as the victim model). Each pair is rated for **semantic similarity (1–5 Likert scale)** by **five human experts and GPT-5.4**. We then compute **Spearman’s ρ** between these ratings and our embedding-based similarity score.
>
> Human vs. Embed.|LLM vs. Embed.|Human vs. LLM|Human IAA
> -|-|-|-
> 0.73|0.75|0.79|0.84
>
> The results suggest that the embedding similarity provides a reliable proxy. We will include the discussion in the revision.
>
> ---
>
> > **[W3,Q2]** The actual API cost inflation.
>
> **A3:** Thanks for the suggestion. In the table below, using GPT-4o-mini pricing (0.15 USD/M input tokens, 0.60 USD/M completion tokens), we convert token usage into **API cost**, observing that the attack increases both **query and completion tokens per sample**, resulting in an **overall cost increase of about 2.0×–3.0×**. We will include the results in the revision.
>
> Dataset|Query Tokens (Orig.)|Query Tokens (Attack)|Completion Tokens (Orig.)|Completion Tokens (Attack)|USD Cost
> -|-|-|-|-|-
> VQAv2|~120k|~403k|~3k|~11k|**3.4×**
> SciFiBench|~81k|~231k|~3k|~7k|**2.8×**
> CLEVR-Math|~134k|~274k|~5k|~13k|**2.1×**
>
> ---
>
> > **[W4]** The range and mapping of the rewards.
>
> **A4:** Thanks for the comment. In practice, the two reward components work stably (see Fig. 5). Moreover, they are **not used in isolation**: both scores, together with the **history buffer**, are fed to the **quality judge**, forming a **multi-signal feedback loop** to guide subsequent rewrites and stabilizes the iterative process. We will clarify this in the revision.
>
> ---
>
> > **[W5]** Include samples that already hit the tool cap.
>
> **A5:** Thanks for the suggestion. Here, we recompute ΔSteps by aggregating both **cap-hit and non-cap-hit samples** (W. Agg.), and report the results below together with the original results (W/O. Agg.).
>
> Aggregation (Low-Budg.)|Qwen2-VL-7B|GPT-4o-mini
> -|-|-
> ΔSteps (W/O. Agg.)|3.33±0.99|2.13±0.54
> ΔSteps (W. Agg.)|3.12±0.73|1.97±0.32
>
> We observe that such modification results in **minor changes in ΔSteps** and the overall conclusion remains consistent.
>
> ---
>
> > **[W6,Q3]** Consider a Zero-Shot baseline and Cover DoS works on reasoning Models.
>
> **A6:** Thanks for pointing this out. We will **revise the Related Work section to include these works**. Following your suggestion, we perform a preliminary comparison with **ThinkTrap** and **Zero-Shot rewriting**, with results summarized below.
>
> Method|ΔSteps↑|\|Similarity\|↓
> -|-|-
> Zero-Shot|1.84±0.81|2.87±0.62
> ThinkTrap|2.31±0.83|2.12±0.47
> **STA (ours)**|**3.33±0.99**|**0.70±0.20**
>
> Both baselines increase steps but remain weaker than STA. Moreover, their design of **explicitly encouraging prompt complexity** leads to substantially bad semantic quality, making the attack **less stealthy and easier to detect**.
>
> ---
>
> > **[Q4]** Results when explicit tool names are hidden.
>
> **A7:** Thanks for the interesting comment. We replace the tool name with generic identifiers (e.g., tool 1, tool 2, ...) for comparison here (Qwen-2-7B, Low-Budg.).
>
> Method|ΔSteps↑|\|Similarity\|↓
> -|-|-
> Hidden name|3.18±0.95|0.76±0.18
> **STA (ours)**|**3.33±0.99**|**0.70±0.20**
>
> The results show that this only causes **a minor performance change**, indicating that the learned policies do not overfit to specific tool interfaces.
>
> ---
>
> > **[L1]** Add discussion on negative societal impact.
>
> Thanks for the suggestion. We will expand negative societal impact in the Impact Statement.
>
> ---
>
> Thanks again for the constructive feedback. If reviewer x2yo has any remaining concerns, we are happy to clarify further.

---

### Official Review · Reviewer_zD9U · 2026-03-13

**Soundness:** 3
**Presentation:** 2
**Significance:** 3
**Originality:** 2
**Overall Recommendation:** 4
**Confidence:** 3

**Summary:**

The paper introduces Sponge Tool Attack (STA), a new attack targeting tool-augmented LLM agents. The authors argue that most work on agentic systems focuses on improving reasoning ability and tool integration, while the security implications of tool-calling pipelines remain underexplored. In particular, they identify a new attack surface termed Denial-of-Efficiency (DoE), where an attacker does not alter the final answer but instead forces the agent to perform unnecessarily long and inefficient reasoning trajectories, increasing computational cost.

STA operates under a query-only threat model, where the attacker can only modify the input prompt without access to the underlying model or tools. The attack rewrites the user prompt in a way that preserves semantics but encourages the model to perform excessive tool calls and verbose reasoning steps before producing the correct answer. The framework uses an iterative multi-agent rewriting pipeline that generates semantically faithful prompt variants while inducing inefficient reasoning behavior. The authors evaluate STA across several models, tools, datasets, and agent frameworks, and show that the attack consistently increases the number of tool-calling steps and overall computational overhead while maintaining task correctness.

**Compliance With Llm Reviewing Policy:**

Affirmed.

**Final Justification:**

Overall, the rebuttal provides additional empirical evidence, clearer mechanism-level intuition, and stronger validation across components and model scales, and combined with the strength of the core Denial-of-Efficiency idea, this sufficiently addresses several of my initial concerns and justifies a higher score.

**Key Questions For Authors:**

**Attack mechanism analysis.**
What specific reasoning patterns introduced by STA cause agents to increase tool calls? Can the authors provide qualitative trajectory examples showing how the rewritten prompts lead to inefficient reasoning?

**Component contribution.**
Which components of the STA framework contribute most to the attack effectiveness? Can the authors provide ablations on the multi-agent rewriting pipeline and policy induction steps?

**Comparison with simpler baselines.**
How does STA compare to simple prompt injections that encourage redundant verification or multiple reasoning paths? Would such simple baselines already produce similar Denial-of-Efficiency effects?

**Effectiveness on larger models.**
The experiments focus mainly on relatively small or mid-sized models. How effective is STA on larger frontier models with stronger reasoning and tool-use capabilities?

**Limitations:**

Yes

**Strengths And Weaknesses:**

**Strengths**

1. The paper highlights a relevant and underexplored attack surface in tool-augmented LLM agents, focusing on efficiency degradation rather than correctness manipulation.

2. The Denial-of-Efficiency (DoE) concept is well motivated. In many real-world settings users interact with models through APIs where cost depends on token usage and tool calls, making efficiency attacks practically relevant.

3. The proposed Sponge Tool Attack (STA) framework provides a concrete mechanism to exploit this attack surface through prompt rewriting under strict query-only access.

4. The empirical evaluation covers multiple models, tools, agent frameworks, and datasets, providing evidence that the attack generalizes across settings.

5. The results show that STA can significantly increase tool-calling steps and reasoning verbosity while preserving the final answer, which supports the central claim.

**Weaknesses**

1. The paper lacks qualitative analysis of the attack behavior. It would be useful to understand what specific reasoning patterns or prompt modifications cause the agent to generate longer trajectories.

2. The experiments do not include larger frontier models, making it unclear whether the attack remains effective on stronger models.

3. The paper lacks component-level ablations isolating which parts of the STA framework contribute most to the attack effectiveness.

4. Comparisons against strong but simple baselines are missing. For example, simple prompt injections that encourage redundant verification, multiple reasoning paths, or repeated tool checks might already produce similar inefficiency.

5. The paper focuses mainly on quantitative metrics such as increased tool steps, but does not deeply analyze the mechanisms that cause the agent to behave inefficiently.

6. The attack evaluation emphasizes step counts and efficiency degradation, but does not clearly analyze the tradeoff between stealthiness and attack strength.

---

> ### Author Rebuttal · Authors · 2026-03-30
>
> It’s our great honor to receive Reviewer zD9U's thoughtful comments and kind words to our work. We would like to address the concerns as below.
>
> ---
>
> > **[W1,W5,Q1]** Attack mechanism analysis.
>
> **A1:** Thanks for the comment. We would like to respond from two perspectives: **First**, **Fig. 7** and **Lines 404–418 (right side)** show that STA induces systematic shifts in tool usage, mainly in middle-to-late reasoning stages, indicating amplified error accumulation and increased tool misselection. A concrete prompt-rewrite example is also provided in the **Appendix (page 17)**.
>
> **Second**, we summarize the main tool-call flow of the example on page 17 (the full reasoning trajectory is too long to present here):
>
>     Original:
>     identify uncolored regions→image-caption→check adjacency constraints→count valid colorings
>     Sponged:
>     identify uncolored regions→image-caption→repeated queries on Four Color Theorem→image-caption→check adjacency constraints→count valid colorings
>
> This example shows that STA elongates trajectories by redirecting the agent toward weakly relevant but plausible subroutines. We will add the full trajectory into the revision.
>
> ---
>
> > **[W2,Q4]** Evaluations on larger models.
>
> **A2:** Thanks for the suggestion. We further conduct an experiment on two larger models (72B, 108B). The setting is similar to Table 2 and we report the results under low budget in the table below.
>
> |Victim Model|Δ Steps ↑| \|Similarity\| ↓|Δ Cap Hits (%) ↑|
> |-|-|-|-|
> |QVQ-72B-Preview|2.05±0.22|0.94±0.18|6.12|
> |GLM-4.5V (108B)|1.78±0.17|0.96±0.16|5.46|
>
> As shown above, tool-calling steps increase by \~2.0 on average while preserving high semantic similarity, and the proportion of samples reaching the tool budget remains considerable (\~5%).
>
> ---
>
> > **[W3,Q2]** Component-level ablation study.
>
> **A3:** Thanks for highlighting this consideration. In our paper, **Fig. 6(a)** studies the effect of rewriter and policy. In the appendix, **Table 4(A)** studies the effect of reward model, and **Table 4(B)** studies the utility of the history buffer and judge design.
>
> Here, we further conduct two analyses: **(i) w/o policy inductor**, removing policy induction; **(ii) rewriter-only**, removing both judge feedback and policy induction. Results are shown below (other settings follow Table 4). Both scenarios randomly sample raw trajectories for the guidance of rewriting during attack execution.
>
> |Setting|\|Similarity\| ↓|Δ Cap Hits (%) ↑|
> |-|-|-|
> |Full STA (default)|0.70±0.20|26.54|
> |w/o policy inductor|1.73±0.41|12.08|
> |rewriter-only|2.16±0.54|10.84|
>
> As shown above, removing the **policy inductor** leads to a clear attack degradation, indicating that policy induction plays an important role in **stabilizing and targeting the rewriting behaviors** during attack execution. When further removing the **judge feedback**, the performance decreases further, suggesting that rewriting alone is insufficient to produce effective attacks.
>
> Combined with existing ablations, several observations emerge: **Fig. 6(a)** shows that stronger rewriters and larger probe–policy scales improve attack effectiveness, while **Table 4(A)** suggests the reward model choice has minor impact. In contrast, **Table 4(B)** and our **new ablations** indicate that **policy induction and the history/judge feedback loop are the key components**. Overall, STA’s performance mainly stems from the **interaction between rewriting, feedback-driven exploration, and policy induction (the main loop in Fig. 2)**.
>
> ---
>
> > **[W4,Q3]** Comparison with simpler baselines.
>
> **A4:** Thanks for raising this point. We introduce a prompt-injection (PI) baseline, where the text blow is appended to the original query:
>
> - "Please reason thoroughly: explore multiple paths and perform extra checks before answering."
>
> We use Gemma-3-27B as the victim model and the tool-calling budget is set to 15. The results are as below:
>
> |Setting|Δ Steps ↑|\|Similarity\| ↓|Δ Cap Hits (%) ↑|
> |-|-|-|-|
> |PI Baseline|0.47±0.14|0.52±0.19|0.91|
> |Full STA|1.36±0.50|0.87±0.23|3.83|
>
> We can observe that the baseline slightly increases reasoning steps but is much weaker. Simple injections add surface-level redundancy, whereas STA induces **structured inefficiency** (e.g., targeting hard-to-verify options in multiple-choice questions or subgoals that stall reasoning in open-ended tasks), leading to more pronounced effects.
>
> ---
>
> > **[W6]** Analyze the tradeoff between stealthiness and attack strength.
>
> **A5:** Thanks for the comment. We would like to respectfully clarify that, strictly speaking, stealthiness and attack strength do not exhibit such tradeoff; STA achieves both by design. Performance drops mainly when key components are removed (see Table 4 and our response A3) or modified (see Fig. 6(a)). We will clarify this in the revision.
>
> ---
>
> Thanks again for the thoughtful and constructive feedback. We would definitely love to further interact with the reviewer if there are any remaining questions.

---

> > ### Author Rebuttal · Reviewer_zD9U · 2026-04-05
> >
> > I thank the authors for the detailed and thoughtful rebuttal. The additional analyses and experiments address several of my concerns and improve the paper.
> >
> > The qualitative example and accompanying explanation provide a clearer picture of how STA induces inefficiency by introducing weakly relevant but plausible subroutines, which helps clarify the underlying mechanism. The additional ablations are also useful, and the results make it clearer that policy induction and the feedback loop are key contributors to attack effectiveness. The inclusion of a prompt injection baseline strengthens the empirical evaluation by showing that simple prompt-level strategies are significantly less effective than STA.
> >
> > I also appreciate the additional experiments on larger models. While the absolute degradation is moderate, the results suggest that the attack continues to have an effect even at larger scales, which strengthens the claim of generality.
> >
> > Some concerns remain, particularly around the lack of a more systematic analysis of failure modes and the limited strength of baseline comparisons. The discussion of stealthiness versus attack strength could also be better supported empirically. However, the rebuttal addresses a substantial portion of the original concerns and improves both clarity and empirical support.
> >
> > Overall, I am inclined to increase my score.

---

> > > ### Author Response · Authors · 2026-04-05
> > >
> > > We are greatly encouraged to hear that our rebuttal has addressed a substantial portion of the concerns and improved both the clarity and empirical support of the paper, and we **sincerely appreciate Reviewer zD9U’s support in raising the score to 4**.
> > >
> > > In the revision, **as promised, we will strengthen the preliminary analyses introduced in the rebuttal and provide more comprehensive discussion accordingly**. Specifically, for **failure modes of the vicitm agent**, besides the example provided in A1, we will add more systematic analyses and examples, and provide the full trajectory comparisons in the revision (the full trajectory is too long to present here). Here we further conclude and list some of the observed failure modes that frequently happened:
> > >
> > > - *Weakly-Relevant Knowledge Detour*: plausible but non-productive background exploration.
> > >
> > > - *Tool Misselection Substitute Drift*: detouring into less efficient but still plausible tool choices.
> > >
> > > - *Re-perception Overhead*: repeated perceptual grounding without adding much new information.
> > >
> > > - *Late-Stage Backtracking*: inefficiency emerges mainly after partial progress has already been made.
> > >
> > > For **baseline comparisons**, we will further strengthen this part by including more competitive prompt-only baselines (including the one we've provided in our response A2(iii) to Reviewer 7R3h). For **stealthiness versus attack strength**, beyond the current ablation evidence, we will add more direct empirical support, such as the relationship between ΔSteps bins and semantic preservation, as illustrated in the table below (Gemma-3-27B under low-budget setting):
> > >
> > > |ΔSteps bin|0.0–0.4|0.4–0.8|0.8–1.2|1.2–1.6|1.6–2.0|
> > > |-|-|-|-|-|-|
> > > |Avg.\|Similarity\| ↓|0.85±0.29|0.86±0.24|0.83±0.27|0.88±0.26|0.85±0.21|
> > >
> > > If a clear tradeoff existed, we would expect Avg. |Similarity| to increase (i.e., become less similar) as ΔSteps grows. However, the results do not exhibit such a trend, suggesting that no strict tradeoff is observed in practice. We will include this analysis together with our response in A5 into the revision.
> > >
> > > Thanks again to Reviewer zD9U for **acknowledging our efforts in the rebuttal** and for **the kind support of our work**. We greatly appreciate **the valuable time and effort Reviewer zD9U devoted to reviewing our paper**.

---

### Decision · Program_Chairs · 2026-04-30

**Decision:**

Accept (regular)

**Comment:**

This paper introduces a new attack on tool-augmented LLM agents, namely the Sponge Tool Attack (STA), which targets efficiency rather than correctness. While there was some initial disagreement among reviewers, the rebuttal has successfully addressed the major concerns, and all reviewers are now satisfied and leaning toward acceptance.

Some concerns remain, particularly regarding evaluation depth and baseline comparisons, but these do not affect the overall positive assessment of the paper. The contribution is still considered valuable to the community, and the consensus remains toward acceptance.